# Intermittent Fasting in Breast Cancer: A Systematic Review and Critical Update of Available Studies

**DOI:** 10.3390/nu15030532

**Published:** 2023-01-19

**Authors:** Marios Anemoulis, Antonios Vlastos, Vasileios Kachtsidis, Spyridon N. Karras

**Affiliations:** 1Medical School, Aristotle University, 55535 Thessaloniki, Greece; 2Laboratory of Biological Chemistry, Medical School, Aristotle University, 55535 Thessaloniki, Greece

**Keywords:** intermittent fasting, breast cancer, quality of life, recurrence, chemotherapy, toxicity

## Abstract

Breast cancer (BC) is the most-frequent malignancy amongst women, whereas obesity and excess caloric consumption increase the risk for developing the disease. The objective of this systematic review was to examine the impact of intermittent fasting (IF) on previously diagnosed BC patients, regarding quality of life (QoL) scores during chemotherapy, chemotherapy-induced toxicity, radiological response and BC recurrence, endocrine-related outcomes, as well as IF-induced adverse effects in these populations. A comprehensive search was conducted between 31 December 2010 and 31 October 2022, using the PubMed, CINAHL, Cochrane, Web of Science, and Scopus databases. Two investigators independently performed abstract screenings, full-text screenings, and data extraction, and the Mixed Method Appraisal Tool (MMAT) was used to evaluate the quality of the selected studies. We screened 468 papers, 10 of which were selected for data synthesis. All patients were female adults whose age ranged between 27 and 78 years. Participants in all studies were women diagnosed with BC of one of the following stages: I, II (HER2−/+), III (HER2−/+), IV, LUMINAL-A, LUMINAL-B (HER2−/+). Notably, IF during chemotherapy was found to be feasible, safe and able to relieve chemotherapy-induced adverse effects and cytotoxicity. IF seemed to improve QoL during chemotherapy, through the reduction of fatigue, nausea and headaches, however data were characterized as low quality. IF was found to reduce chemotherapy-induced DNA damage and augmented optimal glycemic regulation, improving serum glucose, insulin, and IGF-1 concentrations. A remarkable heterogeneity of duration of dietary patterns was observed among available studies. In conclusion, we failed to identify any IF-related beneficial effects on the QoL, response after chemotherapy or related symptoms, as well as measures of tumor recurrence in BC patients. We identified a potential beneficial effect of IF on chemotherapy-induced toxicity, based on markers of DNA and leukocyte damage; however, these results were derived from three studies and require further validation. Further studies with appropriate design and larger sample sizes are warranted to elucidate its potential standard incorporation in daily clinical practice.

## 1. Introduction

Breast cancer (BC) is the most-frequent malignancy amongst women, after non-melanoma skin neoplasms [1]. Approximately 25% of estimated new malignancy cases and 14% of estimated neoplasm-induced deaths in women are attributed to BC [1,2]. Previous epidemiological studies identified obesity as a risk factor for BC development and recurrence in BC patients, and strategies for optimal body weight are considered essential in primary and secondary prevention of this clinical entity [2]. Caloric restriction (CR), without malnutrition, has been considered as one the most-effective interventions for cancer prevention in mammals [3].

Continuous energy restriction (CER), in the form of a daily 30% reduction of the basic metabolic rate, while maintaining required amounts of vitamins, minerals, and other necessary nutrients, combined with a less sedentary lifestyle, is a common strategy for weight loss [4,5]. Nevertheless, available studies indicate a moderate degree of adherence within 1–4 months after dietary intervention [4].

In this regard, alternative methods have been suggested as appropriate dietary regimens, including intermittent energy restriction (IER) as an umbrella term that includes two different subtypes of fasting: intermittent fasting (IF) and time-restricted feeding (TRF) [4]. Intermittent fasting (IF) involves short periods of marked energy restriction followed by periods of usual caloric intake [6].

IF dietary patterns consist of long periods (e.g., 16–48 h) of little to no caloric consumption, repeatedly alternating with periods of ad libitum intake (at one’s pleasure, abbreviated to “ad lib”). IF variations include: (i) alternate day fasting (ADF), (ii) alternate day modified fasting (ADF), (iii) fasting 2 days per week (2DW), and (iv) periodic fasting (PF) lasting 2–21 days [4]. The ADF IF-subtype consists of alternating the day when the energy limit is 75%, known as the “fast day”, and the day when food is eaten ad libitum [7]. Time-restricted feeding (TRF) is an IF diet that focuses on eating within a specific time frame, within a day (usually from 6–12 h) [7].

Whilst chronic caloric deficit cannot be practically implemented for cancer patients, short fasting periods could comprise an alternative approach as a candidate adjunctive tool for cancer prevention and treatment. Results from available studies, however, are conflicting, particularly on certain types of malignancy, including BC. This is mainly due to a scarcity of high-quality evidence, despite available preclinical data suggesting beneficial effects of IF in chemotherapy-related toxicity and tumor growth [8]. We aimed to systematically review available evidence regarding the implementation of IF in BC patients, in a secondary prevention setting, with a discourse on current knowledge gaps and the future research agenda. The aim of this systematic review was to explore the effect of intermittent fasting (IF) on already diagnosed BC patients, regarding the QoL during chemotherapy, chemotherapy-induced toxicity, radiological response and BC recurrence, endocrine-related outcomes, and IF-induced adverse effects. This analysis addressed the potential effects of IF implementation on secondary prevention of BC. The rationale behind this analysis is based on a plethora of earlier studies [9,10,11,12,13,14] that demonstrated that short periods of very low caloric intake, including either periods of short-term fasting (2–4 days) or dietary manipulation of specific macronutrients, can be effective at delaying primary tumor growth. Conversely, excess consumption of animal-derived protein is linked with increased cancer risk and all-cause mortality. Different forms of intermittent fasting (IF) and time-restricted feeding (TRF) are broadly characterized by cyclical periods of low caloric intake or complete fasting interspersed between periods of ad libitum (AL) feeding. IF and TRF result in a dramatic reduction in tumor growth and have garnered traction both as an adjuvant to chemotherapy and as a tool for cancer prevention with promising translational applications.

## 2. Methods

### 2.1. Data Sources and Search Strategy

We conducted a systematic review regarding the safety and efficacy of IF among BC patients in the reduction of chemotherapy-related side-effects and the prevention of disease recurrence. A comprehensive literature search was carried out, using the following search strategy. We combined the terms ‘’intermittent fasting”, OR ‘’alternate day fasting”, OR ‘’time-restricted fasting”, OR ‘’fasting’’ to identify fasting component and ‘’breast cancer”, OR ‘’breast malignancy’’ to identify the BC component. The PubMed, CINAHL, Cochrane, Web of Science, and Scopus databases were used as the search platforms. The search was performed in the fields of the article title, abstract, and Medical Sub-Heading (MeSH) terms in PubMed and the fields of the article title, abstract, and keywords in CINAHL and Scopus. Reference lists of the selected articles were searched to identify additional articles. This review protocol was developed, but not registered; therefore, a complete copy of the Systematic Reviews and Meta-Analyses checklist (PRISMA Checklist 2020) is given in the Supporting Information (Appendix A).

### 2.2. Eligibility Criteria

Original studies that were published between 31 December 2010 and 31 October 2022 and investigated IF in BC patients were included. The selected studies either reported the association between IF and BC-related outcomes or provided sufficient data to calculate relevant measures of association. We included cross-sectional, longitudinal, case–control, and cohort studies.

We included outcomes related to quality of life (QoL), chemotherapy-induced toxicity, response after chemotherapy, radiological disease recurrence, adverse effects, and endocrine-related outcomes of IF, in previously diagnosed BC patients. The literature search was limited to articles published in English and solely including BC survivors over the age of 18. Duplicate articles were removed.

We excluded studies not published in English, not performed in humans, as well those performed on a healthy population and that included humans younger than the age of 18. The titles and abstracts of the remaining articles were screened to select publications for the full-paper review. The full papers were then assessed to determine their eligibility using predetermined inclusion and exclusion criteria.

### 2.3. Quality Assessment

We used the Mixed Method Appraisal Tool (MMAT) [15] to evaluate the quality of the selected studies. The MMAT is a critical appraisal tool, designed for the appraisal stage of systematic mixed studies (Appendix A). It allows the appraising of the methodological quality of qualitative research, randomized controlled trials, non-randomized studies, quantitative descriptive studies, and mixed methods studies. The MMAT checks for selection bias, the appropriateness of the measurement for expected exposure, the completeness of outcome data, and the intended exposure. There are five core criteria that are the most relevant to appraise the methodological quality of studies. Each item was rated on a categorical scale (yes, no, and cannot tell); if the details of any quality assessment criteria were not reported in the reviewed papers, each item was categorized as “cannot tell”.

### 2.4. Data Extraction

A template was designed for the data extraction. This included fields for methods (design, sample characteristics, type of IF, exclusion, and inclusion criteria, reported BC-related outcomes, and/or adverse effects of IF of (if reported)). The literature search, title/abstract screening, full-paper assessment, quality assessment of the papers, and data extraction were performed independently by M.A., A.V., and V.K. Any differences in these outcomes were discussed, and the consensus was reached and referred to S.K. for resolution and/or confirmation.

## 3. Results

We identified 468 articles, excluding duplicates. The title/abstract screening excluded 344 articles. In total, 10 out of the 67 remaining articles were included. The reasons for exclusion are given in Figure 1. The selected studies used various study designs, namely cohort (n = 2), clinical trial (n = 1), case series (n = 1), qualitative study (n = 1), controlled cross-over (n = 1), randomized cross-over pilot study (n = 2), randomized controlled observer-blind study (n = 1), and non-randomized cross-over pilot study (n = 1). The sample characteristics are presented in Table 1.

Most studies (n = 7) were hospital-based; two studies recruited participants from both the hospital and the community (n = 2), while one study recruited participants from an unidentified source (n = 1). Sample sizes ranged from 4 to 2413 female patients. All patients were female adults, and their age ranged from 27–78 years. The participants in all studies were women diagnosed with BC of the following stages: I, II (HER2−/+), III (HER2−/+), IV, LUMINAL-A, LUMINAL-B (HER2−/+).

### 3.1. Health Outcomes

#### 3.1.1. Quality of Life: Chemotherapy-Induced Side Effects

A positive effect of IF on QoL scores of BC patients undergoing chemotherapy was reported in four studies. The QoL was assessed through The Functional Assessment Of Cancer Therapy-General (FACT-G), Functional Assessment of Chronic Illness Therapy (FACIT-F), the Big Five Inventory (BFI) scale, as well as the score based on Common Terminology Criteria for Adverse Events of The National Cancer Institute [16]. Overall, BC patients displayed higher tolerance to chemotherapy and less chemotherapy-induced side-effects whilst following an IF regimen.

In detail, Kleckner et al. assessed 4–60 months post-cancer treatment BC survivors, who already reported a fatigue level ≥ 3 on a scale between 0 and 10 [17]. Women followed a 2-week 14: 10 h TRF dietary regimen (this dietary pattern includes 14 h of fasting within the same day) with no inclusion of a control group in the study [17]. Fatigue symptoms were assessed using the FACIT-F and BFI scales pre- and post-intervention. The FACIT-F is divided into five subscales: physical well-being, social wellbeing, emotional well-being, functional well-being, and fatigue [16]. The BFI is a self-reported scale that is designed to measure the major personality traits (extraversion, agreeableness, conscientiousness, neuroticism, and openness) and is comprised of a 10-item fatigue questionnaire, which is also validated and commonly used in the cancer population [18].

A significant improvement of fatigue scale symptoms was evident, as well as high adherence of BC survivors to this TRF regimen. The authors reported a reduction in fatigue severity after 2 weeks of the TRF regimen [17]. Specifically, fatigue scores improved 5.3 ± 8.1 points on the FACIT-F fatigue subscale (*p* < 0.001, effect size (ES) = 0.55), 30.6 ± 35.9 points for the FACIT-F total score (*p* < 0.001, ES = 0.50), and −1.0 ± 1.7 points on the BFI scale (*p* < 0.001, ES = −0.58) [17].

Bauersfeld et al. (2018) focused on the effects of IF on the impact of chemotherapy (six cycles) on BC patients who started fasting 36 h before chemotherapy and stopped fasting 24 h post-chemotherapy (60 h fasting period) [19]. Overall, 34 BC patients were instructed to follow an IF dietary pattern, in the first half of the chemotherapy cycles, followed by a normocaloric diet (Group A; n = 18) or vice versa (Group B; n = 16) [19]. The authors assessed the QoL scores through the Functional Assessment of Cancer Therapy-General (FACT-G) [16]. This is a 27-item questionnaire assessing health-related quality of life, as far as physical, social, emotional, and functional welfare is concerned [16]. Chemotherapy-induced reduction of the QoL was less than the minimally important difference (MID; FACT-G = 5) with short-term fasting (STF), but greater than the MID for non-fasting periods, during Chemotherapy Cycles c1–c3 in comparison to Chemotherapy Cycles c4–c6 [19]. Finally, IF improved chemotherapy-induced fatigue, weakness, and gastrointestinal side-effects [19].

Badar et al. [20] recruited four BC patients at Stage IIB/IIIB/IV treated with docetaxel. In their study, Ramadan-fasting patients firstly received chemotherapy (20 min after sunset) and, then, continued their fasting routine for the rest of the month. During the fasting period, the patients fasted daily from dawn to sunset and permitted food access from sunset to dawn. For a minimum of 2 weeks of “wash out” after the end of Ramadan, patients underwent the same chemotherapy while not fasting.

All patients were monitored daily by phone, regarding chemotherapy-induced adverse effects (assessed through the score-based Common Terminology Criteria for Adverse Events of National Cancer), and a complete blood count, as well as renal and liver function assessment were performed once a week. A total of 12.5% of patients noted nausea improvement during fasting; 50% reported fatigue improvement during fasting; in 62.5% of patients, fasting made them feel better as was demonstrated by The Common Terminology Criteria for Adverse Events of The National Cancer Institute Questionnaire [20].

Mas et al. included 16 semi-structured interviews of BC patients who had been previously (within the last year) treated with chemotherapy [21]. Participants were not instructed according to a specific dietary regimen, but rather, followed nutritional advice from healthcare general practitioners. Patients mainly fasted in order to alleviate chemotherapeutic adverse effects, as well as to cope with treatment-induced anxiety, through obtaining a sense of control over their remedy [21]. The authors reported that fasting improved nausea and vomiting, as well as appetite, satiation, and fatigue between chemotherapy sessions [21]. A case series conducted by Safdie et al. [22] reported on the side-effects of fasting among four BC patients, by utilizing a non-validated questionnaire that used items from the Common Terminology Criteria for Adverse Events of The National Cancer Institute. The authors revealed that nausea, vomiting, diarrhea, abdominal cramps, and mucositis were not reported during fasting and fatigue and weakness were reduced [22]. Finally, Marinac et al. reported that patients undergoing an FMD manifested a greater duration of night sleep for more hours per night compared to those who followed a normocaloric diet (β = 0.20; 95% CI, 0.14–0.26) [23].

#### 3.1.2. Chemotherapy-Induced Toxicity

A positive effect of IF on chemotherapy-induced toxicity was reported in 4 out of 10 studies.

Deoxyribonucleic acid (DNA) damage was evaluated through the COMET assay, which was used for quantification of leukocytic oxidative stress and γ-H2AX phosphorylation (formed by the phosphorylation of the Ser-139 residue of the histone variant H2AX) [24] as a marker of chemotherapy-induced double-stranded DNA breaks. Frequency and toxicity scores were used for the evaluation of individual toxicities.

Zorn et al. studied patients subjected to a minimum of four cycles of chemotherapy, and they fasted for 96 h during half of their chemotherapy cycles and followed a normocaloric diet during the rest of the chemotherapy cycles. IF (periodic fasting subtype STF) was found to significantly moderate (*p* = 0.023) the frequency and severity score of stomatitis (*p* = 0.013), headaches (*p* = 0.002), weakness (*p* = 0.024), and the total toxicity score [25]. Moreover, after the implementation of IF, there was chemotherapeutic endurance improvement (*p* = 0.034), and chemotherapy was notably less frequently postponed [21]. Groot et al. randomized 131 patients with HER2-negative Stage II/III breast cancer, without previously diagnosed diabetes and BMI ≥ 18 kg m^2^, to receive either a fasting mimicking diet (FMD) or their regular diet for 3 days prior to and during neoadjuvant chemotherapy. Moreover, they reported that an ad libitum dietary pattern significantly increased CD45 + CD3 + T-lymphocytic DNA damage post-chemotherapy, in comparison to patients who underwent an FMD (*p* = 0.045) [26].

In a previous study by Groot et al. [27], 13 HER2-negative BC patients at Stage II/III were recruited, of whom 7 were randomized to IF (fasting 24 h before and after neoadjuvant TAC regimen: docetaxel/doxorubicin/cyclophosphamide). In general, IF was well tolerated, whereas the authors reported remarkably higher mean erythrocyte and thrombocyte concentrations 7 days post-chemotherapy, in patients undergoing IF compared with those randomized to the non-IF group (95% CI, *p* = 0.007 and 95% CI, *p* = 0.00007, respectively) [27]. Moreover, in patients adhering to IF, protection from chemotherapy-attributed bone marrow and cellular toxicity was evident, as well as improvement in DNA damage in peripheral blood mononuclear cells (PBMCs), quantified by the level of γ-H2AX analyzed by flow cytometry [27], particularly at 30 min post-chemotherapy.

At baseline (upon chemotherapy initiation), γ-H2AX phosphorylation increased in both the IF and non-IF groups, but 7 days after chemotherapy, it solely declined in the IF group [27].

Dorff et al. [28] studied three cohorts, and they evaluated leukocytic oxidative stress through the COMET assay (single-cell gel electrophoresis) and peripheral blood mononuclear cells. DNA damage increased in all cohorts after chemotherapy; however, there was a decrease in Olive tail moment (OTM) (describes heterogeneity within a cell population and can pick up variations in DNA distribution within the tail) in the 48 h and the 72 h cohorts (range 0.9–20.7) [28]. A positive effect of IF on platinum-induced toxicity (fasting for ≥48 h (*p*  =  0.08)) in minimizing DNA damage in leukocytes [28] was also reported. It was concluded that peri-chemotherapeutic fasting for 72 h is safe and achievable for BC patients [28].

#### 3.1.3. Chemotherapeutic or Radiological Response/Tumor Recurrence

We identified two studies that reported results on therapeutic response and tumor recurrence.

De Groot et al. [26] studied the effects of and FMD on tumor growth and response according to the Miller and Payne scores (a grading system comparing tumor cellularity between pre-neoadjuvant core biopsy and the definitive surgical specimen). A radiologically complete or partial response was found to occur more regularly in patients using the FMD (OR 3.168, *p* = 0.039). Furthermore, per-protocol analysis showed that the Miller and Payne 4/5 pathological response, indicating 90–100% tumor cell loss, was more likely to occur in patients using the FMD (OR 4.109, *p* = 0.016) [26]. Marinac et al. (2016) utilized data derived from 2413 women with early-stage invasive BC (semiannual telephone calls) and studied tumor recurrence, during a mean of 7.3 years [23]. Overnight fasting for ≤13 h revealed a higher risk of BC mortality (hazards ratio, 1.21; 95% CI, 0.91–1.60) or a higher risk of all-cause mortality (hazards ratio, 1.22; 95% CI, 0.95–1.56) [23]. It should be emphasized that fasting per night ≤ 13 h, was associated with a 36% higher risk for BC recurrence, in contrast to fasting ≥ 13 h (hazards ratio, 1.36; 95% CI, 1.05–1.76) [23].

#### 3.1.4. Endocrine-Related Outcomes

We identified four studies that reported data on insulin, glucose, ketones, insulin-like growth factor-1 (IGF-1), and IGF binding protein (IGFBP) concentrations.

Zorn et al. [25] studied the effects of IF on serum metabolic parameters before each chemotherapy cycle. Metabolic parameters included insulin, IGF-1, thyroid-stimulating hormone (TSH), free triiodothyronine (fT3), and free thyroxine (fT4) serum concentrations. The authors reported a significant decrease of mean fT3 concentrations in the STF group (−0.47 ± 0.09; 95% CI, 0.64–(−0.30); *p* < 0.001), while mean fT4 increased significantly during the STF cycles compared to the normocaloric cycles (0.82 ± 0.37; 95% CI, 0.09–1.55; *p* = 0.028). Finally, both mean insulin (−169.4 ± 44.1; 95% CI, 257.1–(−81.8); *p* < 0.001) and IGF-1 concentrations (−33.3 ± 5.4; 95% CI, 44.1–(−22.5); *p* < 0.001) significantly decreased during the STF cycles [25].

De Groot et al. (2015) [27] studied serum glucose, insulin, IGF-1, thyroid-stimulating hormone (TSH), and insulin growth factor binding protein 3 (IGF-BP3) concentrations, in both IF and non-IF (controls). Venous blood samples were drawn before randomization, at a maximum of 2 weeks prior to treatment (baseline) and directly before chemotherapy administration. Median blood glucose values increased in both groups, between the two time points (*p* = 0.042 and *p* = 0.043, respectively).

In the STF group, no significant difference in median insulin concentrations between the two time points was reported, but in the non-STF group, fasting insulin was increased (*p* = 0.043). Mean IGF-1 concentrations increased (*p* = 0.012) in the STF group (who fasted 24 h before and after commencing chemotherapy), but no change was reported in the non-STF group, whereas no change was observed in any group for IGF-BP3 concentrations. Finally, TSH significantly decreased (*p* = 0.034) in the non-STF group, but not in the STF group [27].

Dorff et al. [28] studied three cohorts who fasted before chemotherapy for 24, 48, and 72 h (divided into 48 pre-chemo and 24 post-chemo) in BC patients who had been previously treated with TCH (docetaxel, carboplatin, trastuzumab). Of major interest, IGF-1 concentrations decreased by a mean of −30% (−44%, −12%) in the 24 h cohort, −33% in the 48 h cohort, and −8% in the 72 h cohort 1-day post-chemotherapy (*p* = 0.32 comparing all 3 cohort groups), whereas serum β-hydroxybutyrate concentrations increased in the 48 and 72 h cohorts, post-chemotherapy [28]. Marinac et al. also reported that each 2 h increase in nightly fasting duration was associated with a 0.37 mmol/mol lower hemoglobin A1C (HbA1c) level (β = −0.37; 95% CI, −0.72 to −0.01) [23].

#### 3.1.5. Adverse Effects of IF

Adverse fasting-related effects were reported in four studies. In the vast majority, adverse effects occurred upon initiation of chemotherapy [19], including headaches, fever, insomnia, fatigue, dizziness, lightheadedness, weight loss, hypoglycemia, hyponatremia and hypotension, hunger, slight nausea after intake of broth or juices, and orthostatic reactions, while in some cases, malnutrition and undernutrition were reported [19,20,28]. No severe fasting-related adverse effects were reported, and fasting-induced weight loss was quickly regained by most patients.

## 4. Discussion

To our knowledge, this is the first systematic review on the effects of IF subtypes on BC patients.

We divided the available data into patient-reported symptoms through the QoL scores, as well as clinical and objective serum markers of chemotherapy-induced toxicity and endocrine-related outcomes, in order to identify any potential clinical benefit of IF in the daily clinical setting. The available results were characterized by a high degree of heterogeneity regarding IF regimens, duration, and pre-specified health outcomes in conjunction with the timing of IF implementation, as well as the lack of the inclusion of control groups in most of the studies (n = 7). However, we postulate that the improvement in laboratory markers of chemotherapy-induced toxicity identified in three studies could comprise a useful tool for future well-designed controlled trials in the field.

Most studies (n = 4) that focused on the potential effects of IF on the QoL of BC patients undergoing chemotherapy reported an improvement in the FACT-G and FACIT-F scales, as well as the score-based Common Terminology Criteria for Adverse Events of The National Cancer Institute [16,17,18,19]. However, no additional validated measures of the QoL were incorporated, as well as any adjustments for additional factors that could interfere with patient-reported symptoms. Although BC patients displayed higher tolerance to chemotherapy and manifested an improvement in chemotherapy-induced side-effects (such as fatigue, nausea, vomiting, appetite loss, and anxiety), we consider the available data scarce and of low quality for improving the QoL of BC patients.

In our analysis, we identified an improvement in the markers of DNA damage after chemotherapy, outlining a practical approach for identifying clear benefits through future well-designed trials.

Improvement in leukocytic oxidative stress and γ-H2AX (formed by the phosphorylation of the Ser-139 residue of the histone variant H2AX) [24], as well as Olive tail moment as markers of chemotherapy-induced double-stranded DNA breaks. These markers are extensively used to measure DNA damage post-irradiation, where the expression has been proven to be related to healthy tissue damage [27]. γ-H2AX phosphorylation denotes double-strand DNA breaks’ presence and could be, therefore, used as a marker for chemotherapy-induced toxicity in healthy cells, as observed in a phase I/II trial with patients treated with a combination of chemotherapy and belinostat [27]. Nevertheless, the use of γ-H2AX as a marker for chemotherapy-induced toxicity to normal cells is relatively uninvestigated [27].

We failed to identify significant effects of IF on chemotherapeutic or radiological response and tumor recurrence. The available results were limited and have not incorporated validated measures of disease recurrence, as well as in one case based on self-reported data and semi-annual telephone calls.

These results could not confirm previous results in mouse models, where two fasting cycles in combination with cyclophosphamide were found to be sufficient to slow down tumor growth in the short-term period [29].

It could be hypothesized that, in humans, a higher number of FMD cycles might be even more important to observe some clinical benefit. In that context, improving patient adherence, for longer periods, could be useful to observe any potential FMD-related antitumor activity [29]. Another interesting finding was a reported decrease of insulin and IGF-1 concentrations in the fasting groups in two studies, as well a rise in serum β-hydroxybutyrate concentrations after 48 and 72 h, post-chemotherapy; however, these findings were not adjusted to body weight, age, and muscle mass. Previous animal models indicate that serum β-hydroxybutyrate is a potent endogenous histone deacetylase inhibitor, protecting cells from oxidative stress. On the other hand, increased IGF-1 concentrations inhibit apoptosis, boost cell proliferation, and cause genetic instability, augmenting tumorigenesis [30]. Future studies are required in order to establish a cause and effect association between IF-induced ketogenesis and the effects on the IGF-1 axis and specific BC-related outcomes in humans.

In general, IF was well tolerated in the populations included in this analysis, highlighting that it could be a feasible and safe approach for improving health outcomes in conjunction with validated chemotherapeutic regimens or other treatment modalities. One of the included studies [23] reported that patients who fasted ≥13-h had a decreased risk for BC recurrence, but these findings were not confirmed in other available studies. It should be emphasized that fasting-induced weight loss was abolished after the cessation of IF periods, a finding that could be considered important for patients with cachexia, but also posing the challenge of weight regain in obese women with the disease. Currently, there are four undergoing clinical trials (one was withdrawn) (Table 2) studying the effects of IF in BC patients, which highlights the changing spectrum in the field and the need for well-designed interventional controlled trials.

This systematic review has several limitations. First, we included only studies published in English. Second, due to the heterogeneity with regard to the cancer types, interventions, and endpoints of the included studies, we could not conduct a meta-analysis. Finally, although we contacted the authors requesting additional data for additional outcomes, responses were only limited to those presented in the current analysis, thus additional health outcomes were not included.

As far the strengths of our systematic review are concerned, the included studies contain several types of fasting, providing thorough insight into the wide spectrum of IF subtypes. Additionally, the subject of this review falls under a quite unexplored field of high interest, therefore making our study, from our point of view, innovative.

In conclusion, we failed to identify any IF-related beneficial effects on the QoL, response after chemotherapy, or related symptoms, as well as measures of tumor recurrence in BC patients. We identified a potential beneficial effect of IF on chemotherapy-induced toxicity, based on markers of DNA and leukocyte damage; however, these results were derived from three studies and require further validation.

## Figures and Tables

**Figure 1 nutrients-15-00532-f001:**
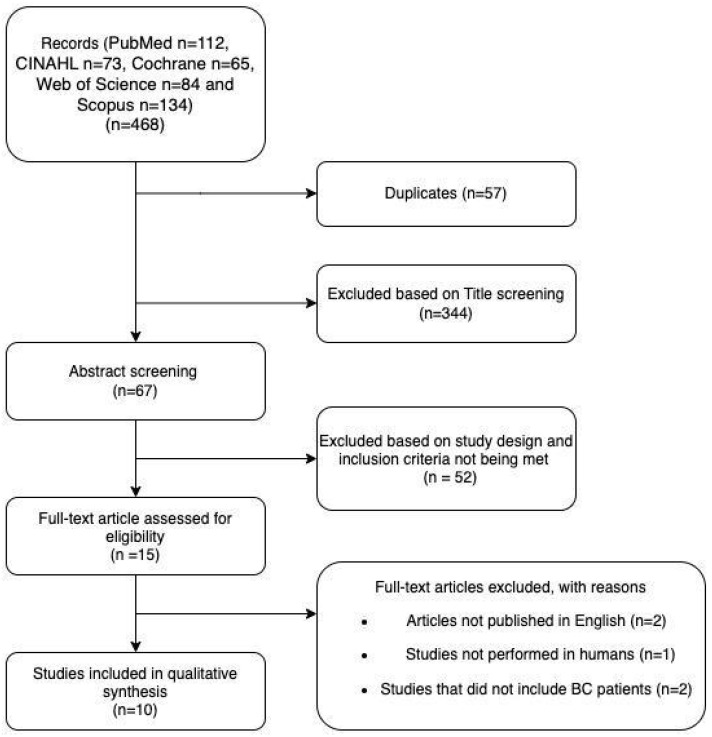
PRISMA flow diagram of the literature review strategy.

**Table 1 nutrients-15-00532-t001:** Effects of intermittent fasting on breast cancer patients.

Author	Year	Population	Fasting Regimen	Results	Type of Study
Kleckner et al. [17]	2021	21 (20 breast cancer survivors)	Women followed a 2-week 14: 10 h TRF dietary regimen (this dietary pattern included 14 h of fasting within the same day) with no inclusion of a control group in the study.	Fatigue scores improved in 2 weeks 5.3 ± 8.1 points on the FACIT-F fatigue subscale (*p* < 0.001, effect size (ES) = 0.55), 30.6 ± 35.9 points for the FACIT-F total score (*p* < 0.001, ES = 0.50), and 1.0 ± 1.7 points on the BFI (*p* < 0.001, ES =−0.58).	Clinical trial
De Groot et al. [26]	2020	131 (HER-2 negative stage II/III breast cancer)	Fasting mimicking diet 3 days before and during neoadjuvant chemotherapy.	There were no differences in Grade 3/4 toxicity during chemotherapy between participants in the fasting and usual care groups. Miller–Payne scores 4/5 pathologic response is more common in FMD. Patients who followed FMD more closely had a more significant percentage of Miller–Payne 4/5 scores.	Randomized, controlled, observer-blind study
Zorn et al. [25]	2020	30 (cancer patients)	96 h fasting for half of scheduled chemotherapy cycles, followed by a regular diet for the remaining cycles.	Fasting is linked to increased serum ketone and lower insulin and IGF-1 level concentrations. The frequency and severity score of stomatitis, headaches, weakness, and the total toxicities’ score were significantly reduced (−10.36 ± 4.44; 95% CI, 19.22–(−1.50); *p* = 0.023) in the STF group. Significantly fewer chemotherapy postponements post-STF were observed, reflecting improved tolerance of chemotherapy.	Controlled cross-over pilot study
Mas et al. [21]	2019	16 (breast cancer)	Participants were not instructed according to a specific dietary regimen, but rather, followed nutrition advice from healthcare practitioners.	Patients fasted in order to alleviate chemotherapeutic adverse effects and treatment-induced anxiety. The authors reported that fasting improved nausea and vomiting, as well as appetite, satiation, and fatigue between chemotherapy sessions.	Qualitative study
Bauersfeld et al. [19]	2018	34 (breast and ovarian cancer)	Patients were randomly assigned to either a short-term fasting diet followed by a normal caloric diet or a normal caloric diet followed by a short-term fasting diet in the first half of chemotherapy.	Within 8 h after treatment, patients on the fasting diet reported improved quality of life and tiredness.	Randomized cross-over pilot study
Dorff et al. [28]	2016	5 (BC female)	Fasting for 24 h, 48 h, or 72 h before chemotherapy.	Fasting is doable and safe. Fasting cohorts had lower incidences of neutropenia and neuropathy. The COMET assay indicated reduced DNA damage in leukocytes from subjects who fasted for ≥48 h (*p* = 0.08). IGF-1 levels decreased by 30, 33, and 8% in the 24, 48, and 72 h fasting cohorts, respectively, after the first fasting period.	Cohort
Marinac et al. [23]	2015	2413 (BC female)	Dietary recalls were utilized to calculate the length of time spent fasting at night.	Fasting for less than 13 h per night is connected with a 36% greater risk of the recurrence of breast cancer when compared to individuals fasting for ≥13 h per night.	Cohort
De Groot et al. [27]	2015	13 (HER2-negative, Stages II/III)	Before and after chemotherapy, patients were randomly assigned to either a 24 h fast or a diet that followed appropriate dietary standards.	Fasting was well tolerated. Fasting may aid in the healing of DNA damage caused by neoadjuvant TAC regimen (docetaxel/doxorubicin/cyclophosphamid).	Randomized pilot study
Badar et al. [20]	2014	4 (BC patients at Stages IIB/IIIB/IV)	Ramadan-fasting patients previously received chemotherapy (20 min after sunset) and, then, continued their fasting routine (daily from dawn to sunset and ate from sunset to dawn) for the rest of the month.	12.5% of patients noted nausea improvement during fasting. 50% of patients reported fatigue improvement during fasting. 62.5% of patients felt better while fasting.	Non-randomized, cross-over, pilot study
Safdie et al. [22]	2009	4 (breast cancer)	Fasting before 48–140 h or fasting after 5–56 h of treatment.	Fasting is safe and well tolerated. Fasting has been linked to decreased tiredness, weakness, and gastrointestinal side-effects.	Case series

**Table 2 nutrients-15-00532-t002:** Clinical trials on breast cancer using intermittent fasting.

Clinical Trial	Study Title	Trial Phase	Total Participants	Status	Primary Outcomes	Details
NCT05023967	Metformin and Nightly Fasting in Women with Early Breast Cancer	2	120	Not yet recruiting	Change in pre–post-treatment Ki67 labeling index in invasive breast cancer (IBC) or ductal carcinoma in situ (DCIS) (in the absence of IBC). Difference in post-treatment adjacent DCIS (in the presence of IBC), if present, or intraepithelial neoplasia Ki67 between arms. Frequency of occurrence of dose-limiting toxicity.	Age: 18 years and older Gender: female Study start: 1 April 2022 Study completion: 30 November 2024
NCT05432856 [31]	A randomized controlled trial of the effect of time restricted eating, healthy eating, and reduced sedentary behavior on metabolic health during chemotherapy for early-stage breast cancer	NA	130	Not yet recruiting	Visceral AT and regional AT pools. Metabolic syndrome severity. Cardiovascular risk estimate. Hemoglobin A1c and C-reactive protein. Chemotherapy symptom presence and severity. Safety.	Age: 18 years and older Gender: female Study start: January 2023 Study completion: December 2024
NCT04560439	Diabetes Prevention Program (METFIT) in Reducing Insulin Resistance in Stage I-III Breast Cancer Survivors	NA	25	Recruiting	Fidelity. Retention. Change in insulin resistance (IR).	Age: 18–75 years Gender: female Study start: 15 June 2022 Study completion: 15 June 2023
NCT04691999	The Effect of Intermittent Fasting on Body Composition in Women With Breast Cancer	NA	0	Withdrawn	Adherence to the intermittent fasting program. Change in body fat.	Age: 20–70 years Gender: female Study start: 2 December 2021 Study completion: August 2023
NCT04330339	Prolonged Nightly Fasting in Breast Cancer Survivors	NA	40	Active, not recruiting	Baseline assessment (measurements of weight, height, quality of life, fatigue, mood, levels of physical activity, and blood markers) prior to the intervention and after completion of 12-week intervention.	Age: 18 years and older Gender: female Study start: 24 July 2020 Study completion: 31 August 2021

## Data Availability

The data presented in the study are available upon request from the corresponding author.

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
