# Peer review of "Intermittent Fasting in Breast Cancer: A Systematic Review and Critical Update of Available Studies"

_nutrients, 2023, doi:10.3390/nu15030532_

Round 1

Reviewer 1 Report

The authors conducted a systematic review with the aim of examining the impact of intermittent fasting (IF) on previously diagnosed BC patients. Here are some comments and suggestions for improvement:

1. The authors do not adequately justify the need to carry out this study. Why could IF be used for the primary or secondary prevention of breast cancer?

2. The purpose of the abstract and introduction differ greatly. Authors are advised to write the objective in a more similar and specific way, as it is written it is quite imprecise and open.

3. The authors performed the search in PubMed, CINAHL, Cochrane and Scopus databases. However, a significant database was not made on the Web of Science. Is there a reason the authors didn't consider this one?

4. It is advisable in Figure 1 to indicate the number of articles found in each database and then indicate those that were eliminated due to being duplicates.

5. In Figure 1 it is advisable to indicate the inclusion criteria that the excluded articles did not meet. For example, 15 were selected for full reading, but 5 were excluded. For what reason or what criteria did these articles not meet to be excluded? In results, the authors declare that "Reasons for exclusion are given in Figure 1.", however, these do not appear.

6. Authors are advised that in the methodology when describing the Mixed Method Appraisal Tool (MMAT) they refer to the scores of this instrument.

7. The section titled “Setting and the participants” really corresponds to results and not to methodology. Authors are advised to consider moving this paragraph to the results section.

8. The following appears in the first paragraph of results: “The quality assessment of the selected articles is also given in Figure 1.”. I think this must be a bug. It is not clear to me what the quality of each of the articles included in the bibliographic review is. Where is this reflected?

9. Just as the authors mention the limitations of their study, they should also mention its strengths.

minor bugs

10. It is recommended that you review the punctuation marks, specifically the “.”. On some occasions space appears both before and after these.

11. Figure 1 should be reflected where it is mentioned.

Author Response

Comments to the Author

The authors conducted a systematic review with the aim of examining the impact of intermittent fasting (IF) on previously diagnosed BC patients. Here are some comments and suggestions for improvement:

We thank the reviewer for the time spent evaluating our work and for the very constructive comments made

  1. The authors do not adequately justify the need to carry out this study. Why could IF be used for the primary or secondary prevention of breast cancer?

Answer: This analysis addressed the potential effects of IF implementation on secondary prevention of BC .The rationale behind this analysis is based on a plethora of  earlier studies that demonstrated that short periods of very low caloric intake, including either periods of short-term fasting (2–4 days) or dietary manipulation of specific macronutrients, can be effective at delaying primary tumor growth. Conversely, excess consumption of animal-derived protein is linked with increased cancer risk and all-cause mortality. Different forms of intermittent fasting (IF) and time-restricted feeding (TRF)are broadly characterized by cyclical periods of low caloric intake or complete fasting interspersed between periods of ad libitum (AL) feeding. IF and TRF result in a dramatic reduction in tumor growthand have garnered traction both as an adjuvant to chemotherapy and as a tool for cancer prevention with promising translational applications .We provide selected previous results on the field.

  1. Lv, M., Zhu, X., Wang, H., Wang, F. & Guan, W. Roles of caloric restriction, ketogenic diet and intermittent fasting during initiation, progression and metastasis of cancer in animal models: a systematic review and meta-analysis. PLoS ONE 9, e115147–e115147 (2014).
  2. Levine, M. E. et al. Low protein intake is associated with a major reduction in IGF-1, cancer, and overall mortality in the 65 and younger but not older population. Cell Metab. 19, 407–417 (2014).
  3. Kim, E. J. et al. Dietary fat increases solid tumor growth and metastasis of 4T1 murine mammary carcinoma cells and mortality in obesity-resistant BALB/c mice. Breast Cancer Res. 13, R78 (2011).
  4. Lamming, D. W. et al. Restriction of dietary protein decreases mTORC1 in tumors and somatic tissues of a tumor-bearing mouse xenograft model. Oncotarget 6, 31233–31240 (2015).
  5. Sundaram, S. & Yan, L. Time-restricted feeding mitigates high-fat dietenhanced mammary tumorigenesis in MMTV-PyMT mice. Nutr. Res. 59, 72–79 (2018)
  6. Lee, C. et al. Reduced levels of IGF-I mediate differential protection of normal and cancer cells in response to fasting and improve chemotherapeutic index. Cancer Res. 70, 1564–1572 (2010)
  7. The purpose of the abstract and introduction differ greatly. Authors are advised to write the objective in a more similar and specific way, as it is written it is quite imprecise and open.

Answer: Thank you for the comment. We have specified the aim of our study at the end of the Introduction.

  1. The authors performed the search in PubMed, CINAHL, Cochrane and Scopus databases. However, a significant database was not made on the Web of Science. Is there a reason the authors didn't consider this one?

Answer: We have omitted Web of Science in our initial analysis , since we consider the databases evaluated adequate for data analysis. However, after the reviewer’s comment ,we have conducted relative search in this database as well, which yielded no additional studies.

  1. It is advisable in Figure 1 to indicate the number of articles found in each database and then indicate those that were eliminated due to being duplicates.

Answer: Revised according reviewer’s suggestion

  1. In Figure 1 it is advisable to indicate the inclusion criteria that the excluded articles did not meet. For example, 15 were selected for full reading, but 5 were excluded. For what reason or what criteria did these articles not meet to be excluded? In results, the authors declare that "Reasons for exclusion are given in Figure 1.", however, these do not appear.

Answer: Revised according reviewer’s suggestion

  1. Authors are advised that in the methodology when describing the Mixed Method Appraisal Tool (MMAT) they refer to the scores of this instrument.

Answer: Revised according reviewer’s suggestion .We mention the MMAT scores at the table on page 18.

  1. The section titled “Setting and the participants” really corresponds to results and not to methodology. Authors are advised to consider moving this paragraph to the results section.

Answer: We moved the paragraph to the recommended section.

  1. The following appears in the first paragraph of results: “The quality assessment of the selected articles is also given in Figure 1.” I think this must be a bug. It is not clear to me what the quality of each of the articles included in the bibliographic review is. Where is this reflected?

Answer: Revised according reviewer’s suggestion

  1. Just as the authors mention the limitations of their study, they should also mention its strengths.

Answer: We mention the strengths of our study in the revised version at the end of the discussion.

  1. It is recommended that you review the punctuation marks, specifically the “.”. On some occasions space appears both before and after these.

Answer: Revised according reviewer’s suggestion

  1. Figure 1 should be reflected where it is mentioned.

Answer: We moved the Figure 1 at the correct section.

Reviewer 2 Report

Dear authors, 

thank you for your submission "Intermittent fasting in breast cancer: a systematic review and critical update of available studies" which I was asked to review. Your article is well structured and provides a very detailed description of the methods applied. There are only minor language annotation (i.e. ad lib ist explained twice) and a more detailed section on the clinical outcome (i.e. pCR rates in case of an NAC trial) could in my opinion improve the draft. Also with a range of n=4-2413 patients in 10 studies, the expression 'the majority of studies...' needs to be put into perspective of #of patients.
Also some more current literature  could enhance the discussion beyond breast cancer.

Other than those little remarks I have no concerns regarding a publication. 

Author Response

Reviewer 2

Comments to the Author

Thank you for your submission "Intermittent fasting in breast cancer: a systematic review and critical update of available studies" which I was asked to review. Your article is well structured and provides a very detailed description of the methods applied.

We thank the reviewer for the time spent evaluating our work and for the very constructive comments made

There are only minor language annotation (i.e. ad lib ist explained twice) and a more detailed section on the clinical outcome (i.e. pCR rates in case of an NAC trial) could in my opinion improve the draft.

Answer: Annotations revised according reviewer’s suggestion .We included all the outcomes we could provide by the articles.

Also with a range of n=4-2413 patients in 10 studies, the expression 'the majority of studies...' needs to be put into perspective of #of patients.

Answer: Revised according reviewer’s suggestion .We have written down the reference numbers of these studies that correspond to the data presented at table 1, where study characteristics are presented.

Reviewer 3 Report

Summary

Authors here have performed a systematic meta analysis on the effects of intermittent fasting on different sub types of breast cancer patients. Authors first describe different approaches usually followed for fasting such as calorie restriction, continuous energy restriction, intermittent energy restriction following different hour periods. They have selected to review 10 papers out of 345 identified articles and provide a comprehensive overview of the sample size, subtypes of breast cancer, different QOL reporting system as well as the outcomes in each study. Authors also acknowledge the discrepancies in outcomes and the lack of studies investigating the same. Review is well structured, first of a kind and would be a great resource and a initiating point for the discussion in this area. 

Minor Points 

-manuscript to be reviewed should have line number option / in a word format to be able to be efficiently review the document.

-please keep the font style and size consistent

-font change in last 3 lines of the introduction

Author Response

Reviewer 3

Comments to the Author

Authors here have performed a systematic meta analysis on the effects of intermittent fasting on different sub types of breast cancer patients. Authors first describe different approaches usually followed for fasting such as calorie restriction, continuous energy restriction, intermittent energy restriction following different hour periods. They have selected to review 10 papers out of 345 identified articles and provide a comprehensive overview of the sample size, subtypes of breast cancer, different QOL reporting system as well as the outcomes in each study. Authors also acknowledge the discrepancies in outcomes and the lack of studies investigating the same. Review is well structured, first of a kind and would be a great resource and a initiating point for the discussion in this area.

-manuscript to be reviewed should have line number option / in a word format to be able to be efficiently review the document.

 -font change in last 3 lines of the introduction

-please keep the font style and size consistent

Answer: We thank the reviewer for the time spent evaluating our work and for the very constructive comments made. We have incorporated all suggested changed in the revised version. We failed to provide to provide line numbers maybe due to journal format issues.

Reviewer 3

Comments to the Author

Authors here have performed a systematic meta analysis on the effects of intermittent fasting on different sub types of breast cancer patients. Authors first describe different approaches usually followed for fasting such as calorie restriction, continuous energy restriction, intermittent energy restriction following different hour periods. They have selected to review 10 papers out of 345 identified articles and provide a comprehensive overview of the sample size, subtypes of breast cancer, different QOL reporting system as well as the outcomes in each study. Authors also acknowledge the discrepancies in outcomes and the lack of studies investigating the same. Review is well structured, first of a kind and would be a great resource and a initiating point for the discussion in this area.

-manuscript to be reviewed should have line number option / in a word format to be able to be efficiently review the document.

 -font change in last 3 lines of the introduction

-please keep the font style and size consistent

Answer: We thank the reviewer for the time spent evaluating our work and for the very constructive comments made. We have incorporated all suggested changed in the revised version. We failed to provide to provide line numbers maybe due to journal format issues.

Reviewer 3

Comments to the Author

Authors here have performed a systematic meta analysis on the effects of intermittent fasting on different sub types of breast cancer patients. Authors first describe different approaches usually followed for fasting such as calorie restriction, continuous energy restriction, intermittent energy restriction following different hour periods. They have selected to review 10 papers out of 345 identified articles and provide a comprehensive overview of the sample size, subtypes of breast cancer, different QOL reporting system as well as the outcomes in each study. Authors also acknowledge the discrepancies in outcomes and the lack of studies investigating the same. Review is well structured, first of a kind and would be a great resource and a initiating point for the discussion in this area.

-manuscript to be reviewed should have line number option / in a word format to be able to be efficiently review the document.

 -font change in last 3 lines of the introduction

-please keep the font style and size consistent

Answer: We thank the reviewer for the time spent evaluating our work and for the very constructive comments made. We have incorporated all suggested changed in the revised version. We failed to provide to provide line numbers maybe due to journal format issues.

Round 2

Reviewer 1 Report

Although the authors have tried to address the suggestions provided above, which has allowed the manuscript to be improved, there are a couple of aspects that should be addressed:

In the Cover Letter they justify the need to carry out this work, and provide references in this regard. It is advisable that this is also briefly reflected in the manuscript. In this way, future readers will have the justification that has led to the realization of this work.

Previously, authors were suggested to include Web of Science in the databases consulted. Although they comment that they have done so, nothing has been modified in this regard in Figure 1. The number of articles found in each database and subsequently the number of duplicates has not been indicated either, as suggested. Please, in case you consider that this is not necessary, justify the reason.

It remains unclear how the MMAT is scored. What items does it contain? How is each item scored? What score is necessary for the study to be of high quality?

Author Response

To the Editors of Nutrients,

  We would like to thank you and the reviewers for the time spent evaluating our manuscript entitled: “Intermittent fasting in breast cancer: a systematic review and critical update of available studies”. Taking into account the comments of the reviewers, we have revised the manuscript accordingly (changes made in the original text are highlighted in yellow). You will find below a rebuttal against each point raised, along with a list of submitted changes.      Thank you in advance for your time and consideration.

                                                                             Kind regards,

                                                                            Professor Spyridon Karras

Reviewer 1

Comments to the Author

In the Cover Letter they justify the need to carry out this work, and provide references in this regard. It is advisable that this is also briefly reflected in the manuscript. In this way, future readers will have the justification that has led to the realization of this work.

Answer:We are sorry for the misunderstanding ,as we thought that this part ,was not suggested to be included in the text ,rather than only in the rebuttal letter. We resolved the issue, by showing the need of addressing this topic as well as the appropriate references at the end of the introduction.

 Previously, authors were suggested to include Web of Science in the databases consulted. Although they comment that they have done so, nothing has been modified in this regard in Figure 1. The number of articles found in each database and subsequently the number of duplicates has not been indicated either, as suggested. Please, in case you consider that this is not necessary, justify the reason.

Answer: We have revised the figure according reviewer’s comment.

It remains unclear how the MMAT is scored. What items does it contain? How is each item scored? What score is necessary for the study to be of high quality?

Answer: We provided further information about the MMAT score at the methods of our study.